# Prompt2Poster: Automatically Artistic Chinese Poster Creation from Prompt Only

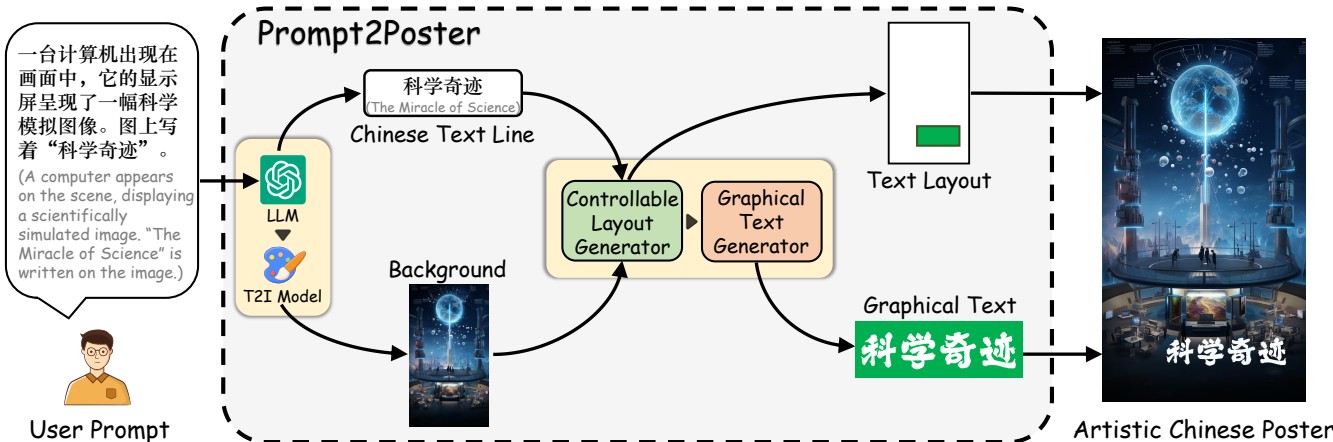

Figure 1: Overview. Prompt2Poster creates artistic Chinese posters with harmonious visual effects and stylized graphical text from a given prompt only. Prompt2Poster first extracts user intention and generates the aligned background with large models, and then creates layouts and graphical texts with designed modules, leading to correct and pleasing visual results.

## ABSTRACT

As a critical component in graphic design, artistic posters are widely applied in the advertising and entertainment industry, thus the automatic poster creation from user-provided prompts has become increasingly desired recently. Although existing Text2Image methods create impressive images aligned with given prompts, they fail to generate ideal artistic posters, especially posters with Chinese texts. To create desired artistic Chinese posters including an aligned background, reasonable layouts, and stylized graphical texts from given prompts only, we propose an automatic poster creation framework, named **Prompt2Poster**. Our framework first utilizes the capacity of the powerful Large Language Model (LLM) to extract user intention from provided prompts and generate the aligned background. For the harmonious layout and graphical text generation, we propose **Controllable Layout Generator (CLG)** and **Graphical Text Generator (GTG)** modules that both leverage sufficient multi-modal information, leading to accurate and pleasurable visual results. Comprehensive experiments demonstrate that our Prompt2Poster achieves superior performance especially on text quality and visual harmony than existing poster creation methods. Our codes will be released after the paper review.

## CCS CONCEPTS

• **Computing methodologies** → *Computer vision tasks*; *Natural language processing*.

## KEYWORDS

Text to Image Generation, Layout Generation, Stylized Font Generation, Large Language Models

## 1 INTRODUCTION

Designing and integrating visually appealing backgrounds and Chinese graphical texts for artistic Chinese posters represent a critical task in visual information presentation. Traditional Chinese poster creation requires professional designers to invest a significant amount of time. In recent years, some methods [10, 22, 31, 32] have been proposed to partly automate the poster creation through multi-step division. However, these methods involve manual operations in some steps, such as selecting existing backgrounds or element attributes, still leading to the necessity of expensive professional labor. Intuitively, obtaining a poster only from a user-provided prompt describing the desired artistic effect would align naturally with human interaction instincts. This significantly streamlines the design process, enhancing creation efficiency, and saving extensive manual labor. Thus, such a prompt-guided automatic poster creation framework has gradually caught greater attention for high interaction, efficiency, and economy.

With the proposal and subsequent developments of Denoising Diffusion Probabilistic Model [3, 12, 25–28], Large Text-to-Image (T2I) Diffusion Models have demonstrated remarkable performance in generating aesthetically pleasing and style-diverse images by user prompts. Some fine-tuning or plus versions of these T2I Diffusion models [3, 12, 25–28], with linguistic and optionally graphical control through LoRA [14] and ControlNet [37], have achieved state-of-the-art for prompt-guided poster generation.

Despite performing excellently in prompt-guided poster generation, limited by the diffusion framework [3, 12, 25–28], these methods face a critical problem of data distribution mixing. For an artistic Chinese poster, the background and Chinese graphical texts

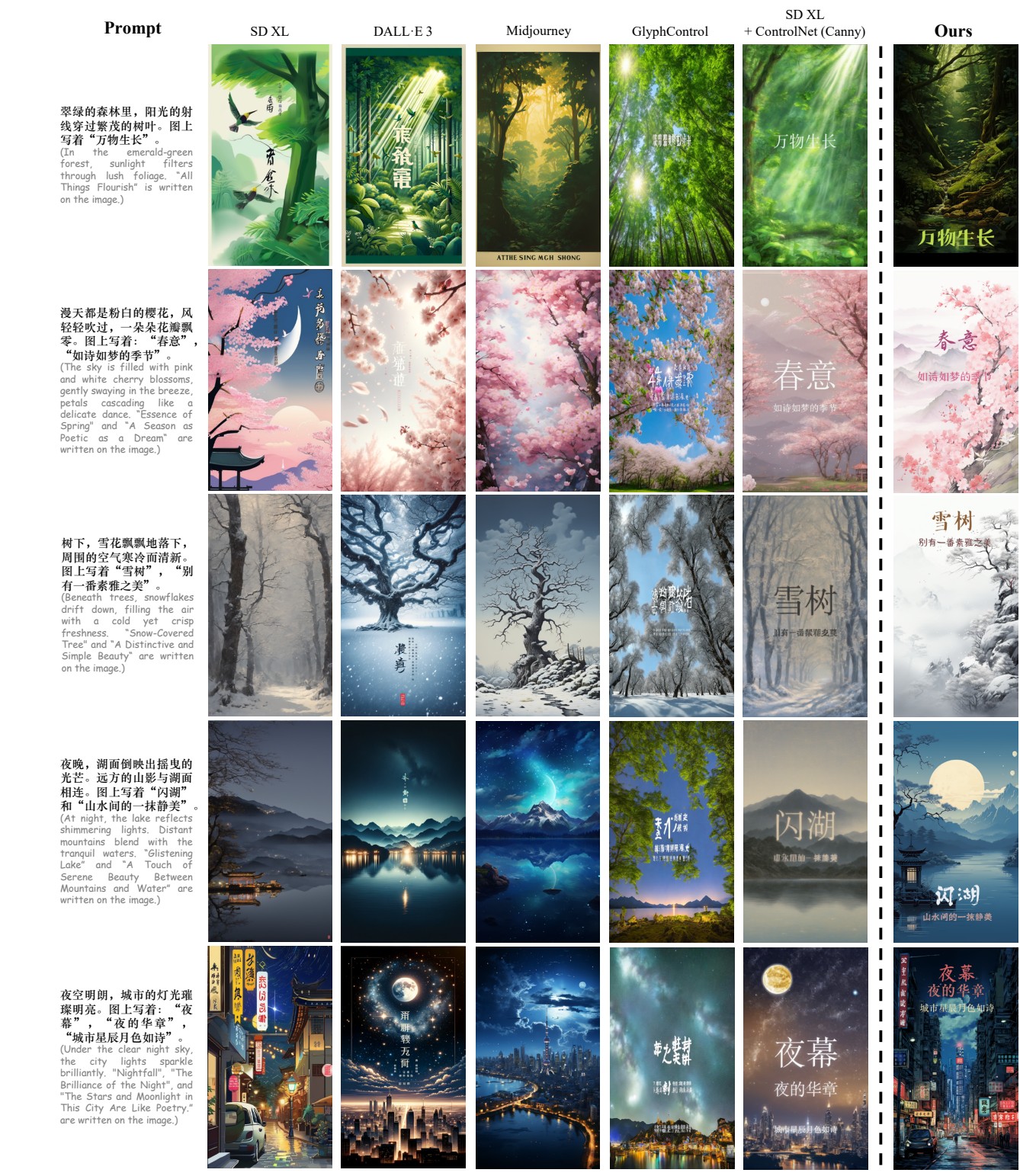

**Figure 2: Artistic Chinese posters generated by our Prompt2Poster and other poster generation methods, including SD XL [25], DALL·E 3 [1], Midjourney, GlyphControl [36], and SD XL + ControlNet (Canny) [37]. The leftmost column represents the prompts, while the remaining six columns display the images generated by different methods. Visual results demonstrate that Prompt2Poster achieves more accurate and pleasurable artistic Chinese poster creation than the compared methods.**

Prompt2Poster: Automatically Artistic Chinese Poster Creation from Prompt Only

ACM MM, 2024, Melbourne, Australia

are extremely different in terms of data distribution. Most of these prompt-guided poster generation methods try to obtain a poster containing the two different distributions from a single distribution, largely weakening their data fitting for the Chinese graphical texts. Thus, they fail to promise the visual correctness and style diversity of Chinese graphical texts on the posters, which are the keys to artistic Chinese poster creation.

To guarantee the fitting of the two different data distributions, we propose a novel poster generation framework, named **Prompt2-Poster**, shown in Fig. 1. According to Fig. 1, although only fed into linguistic control from the user prompt, inner multi-modal information is fully utilized to catch the two different data distributions. Specifically, a Large Language Model (LLM) [20] first extracts various information from the user-provided prompt. Then, a pre-trained Large Text-to-Image (T2I) Diffusion Model with high fitting for the poster background data distribution, generates a high-quality background based on the extracted information. After that, a novel **Controllable Layout Generator (CLG)**, is proposed to harmoniously decide the placement for Chinese graphical texts on the background, based on both the visual and linguistic information, coming from the LLM and T2I model, respectively. The output text layout from CLG contains the placement for Chinese graphical texts. This geometrical text layout, along with previous visual and linguistic information is further sent to a novel proposed **Graphical Text Generator (GTG)**. GTG utilizes the three modal information and continuous spaces to generate correct and style-diverse Chinese graphical texts from the graphic data distribution. A high-quality artistic Chinese poster is conveniently finished after integrating stylized Chinese graphical texts with the background image based on the text layout. As shown in Fig. 2, our Prompt2Poster has a much stronger ability to generate correct and diverse Chinese graphical texts on the poster with a specific prompt, for treating the two different distributions separately. Extensive experiments show the artistic superiority of our Prompt2Poster, CLG, and GTG for artistic Chinese poster generation.

Our contribution can be summarized as follows:

- We introduce Prompt2Poster, a unique framework designed specifically for the automatic creation of artistic Chinese posters. This system fully leverages multi-modal information to accommodate different data distributions.
- A Controllable Text Layout Generator, named CLG, is proposed to produce text layout according to both the background image and Chinese texts, promising harmony between background and Chinese graphical texts.
- We've developed an all-new Graphical Text Generator (GTG) that operates initially in the continuous graphic feature space, aside from predicting color in pixel space. This technique not only maintains text correctness but also enhances the artistic style diversity of Chinese graphical texts.
- Through extensive experimentation, we have been able to demonstrate the effectiveness of our Prompt2Poster, along with its integral modules CLG and GTG, in the creation of artistic Chinese posters. The results highlight the capability of our system to reliably produce high-quality posters, reinforcing the value and potential of our approach.

## 2 RELATED WORK

**Prompt-Guided Poster Generation.** With the rapid development of Large Text-to-Image (T2I) Diffusion Models [3, 12, 25–28], T2I-Diffusion-based poster systems have been widely applied to prompt-guided poster generation for their huge interaction, efficiency and economy. By adding linguistic and optional graphical control by LoRA [14] and ControlNet [37], they return an attractive poster from the noise. However, generating visually correct and various Chinese graphical texts on posters remains a significant challenge for general T2I methods. Recent poster-specific generation methods including AutoPoster [22], Desigen[33], and Visual Layout Composer[29] are proposed, which achieve remarkable creation results for artistic posters. However, their methods most focus on component layout planning thus requiring visual elements as input necessarily, resulting in limitations when users desire to create posters from scratch. We highlight that our Prompt2Poster is a fully automatic and generative framework for artistic Chinese poster creation, which achieves correct and pleasing Chinese posters from given prompts only.

**Controllable Text Layout Generation.** The core of the previous controllable text layout generation models [5, 10, 13, 31, 38] is to predict the placement of graphical texts which is visually harmonious and do not cover the semantic area of the background. Traditional methods [10, 31] take template or optimization strategies while deep-learning-based methods [5, 13, 38] utilize Condition-GAN [9] to achieve better results. However, only receiving the background image, all previous advanced learning-based methods [5, 13, 38] are not able to consider Chinese texts, leading to their complete incompatibility with our Prompt2Poster.

**Stylized Graphical Text Generation.** Following most poster systems [10, 22, 31, 32], in our Prompt2Poster, the stylized attributes of Chinese graphical texts contain the font and color. For font design, previous works primarily concentrate on selecting the most suitable font from a feasible discrete font space, by heuristic method [6] or network learning [22, 31]. The selection in discrete space limits the font variation of Chinese graphical texts. In the realm of color design, some works [15, 31] predict colors based on color rules [16, 30], while the recent TextPainter [8] utilizes a color style encoder to extract color features and use them to reconstruct a visual graphical text.

## 3 METHOD

In this section, we first introduce the overview of our automatic artistic Chinese poster creation framework **Prompt2-Poster** in Sec. 3.1. We then describe the design details of two proposed modules including Controllable Layout Generator (CLG) and Graphical Text Generator (GTG) in the following Sec. 3.2 and Sec. 3.3, respectively. Finally, the loss function and optimization details are given in Sec. 3.4. The overview illustration is shown in Fig .1.

### 3.1 Overview of Prompt2Poster

Given a user prompt $y$, Prompt2Poster aims to automatically create an artistic Chinese poster that satisfies all requirements described by the $y$. Concretely, Prompt2Poster firstly processes the prompt $y$ through a Large Language Model (LLM) [20] to extract the style and content information $s$ of the desired poster background, as well

3

as the Chinese text, set $T$:

$$s, T = \text{LLM}(y). \quad (1)$$

Here, LLM$(\cdot)$ can be various powerful models, such as GPT series[4, 23] and PaLM series [7, 19, 24]. $T$ is the text set containing $M$ lines of Chinese texts which are expected to appear on the poster graphically:

$$T = [t_1, t_2, ..., t_M], \quad t_i = [c_1^i, c_2^i, ..., c_{N_i}^i], \quad (2)$$

where $t_i$ is the $i$-th Chinese text line consisting of $N_i$ Chinese characters defined by the user, and $c_j^i$ is the $j$-th character in the $i$-th text line.

According to the extracted background information $s$, Prompt2-Poster leverages the Large Text-to-Image (T2I) Diffusion Model to create the poster background $I$ with high quality and diversity:

$$I = \text{T2I}(s). \quad (3)$$

We underline that T2I$(\cdot)$ can be various powerful Text-to-Image models, such as Stable Diffusion series [25, 27], DALL·E series [1, 26], or Midjourney, which are high fitting of the data distribution for poster backgrounds.

After obtaining the background $I$, Prompt2Poster generates harmonious and elegant text layout $L = [l_1, l_2, ..., l_M]$ based on the $I$ and $T$. Within this, $l_i = [x_i, y_i, w_i, h_i]$ is the $i$-th layout element, containing the geometrical placement of the $i$-th Chinese text line $t_i$. Specifically, $[x_i, y_i]$ is the center location of the $i$-th Chinese graphical text and $[w_i, h_i]$ is the size. The process of text layout generation can be formulated as:

$$L = \text{CLG}(I, T), \quad (4)$$

where CLG$(\cdot)$ is our designed Controllable Text Layout Generator, which will be further introduced in Sec. 3.2.

With $I, T$ and $L$, we propose a Stylized Graphical Text Generator to create precise and diverse stylized Chinese graphical texts, different from the poster background in terms of data distribution. Each Chinese text line is graphically generated, separately:

$$[g_i, o_i] = \text{GTG}(I, t_i, l_i), \quad (5)$$

where GTG$(\cdot)$ is our Graphical Text Generator introduced in Sec. 3.3, and $g_i, o_i$ are the font and color of the $i$-th Chinese graphical text line, respectively.

Finally, according to the text layout $L$, Prompt2Poster integrates the background image $I$ and Chinese graphical texts together through geometrical pasting. In our Prompt2Poster, although starting from only the linguistic modal $y$, visual and geometrical information $I$ and $L$ also play an important role in the poster generation, which fully utilizes the inner multi-modal information. Through separately generating backgrounds and Chinese graphical texts with different data distributions, Prompt2Poster can obtain overall harmonious and visually appealing artistic Chinese posters with satisfying graphical texts.

### 3.2 Controllable Layout Generator

To guarantee harmony between the background and Chinese graphical texts, how to place the graphical texts on the background elegantly is extremely critical. Serving as a module in our Prompt2-Poster, our Controllable Layout Generator (CLG) determines the placement of Chinese graphical texts by predicting their center

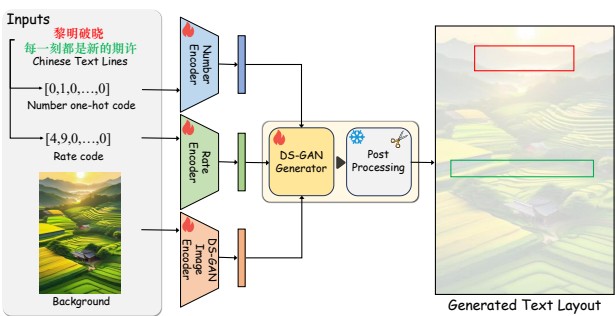

Figure 3: The overview of our Controllable Layout Generator. Trainable NE, RE, and fixed PC are added based on the DS-GAN Generator and DS-GAN Image Encoder.

locations and sizes on the background image. Previous state-of-the-art controllable text layout models [5, 13, 38] are all designed for generating elements only according to the background $I$. To make the text layout generation controlled by both the background and Chinese texts, further acceptance of linguistic information in our CLG is necessary. This is achieved by adding trainable **Number Encoder** (NE), **Rate Encoder** (RE), and fixed **Post-processing Cropping module** (PC) based on an advanced layout model DS-GAN. Besides considering background $I$, CLG promises the text layout $L$ completely matches the Chinese text set $T$ in terms of element number and text length [13].

The generation process of CLG is shown in Fig. 3. A structurally similar discriminator is also designed for training, and more training details are introduced in the supplementary material. Specifically, for a text set $T$ containing $M$ lines of Chinese texts, NE receives an $x$ dimension one-hot layout number vector $H = (h_1, h_2, ..., h_x)$ and RE takes a same dimension rate vector $R = (r_1, r_2, ..., r_x)$. $x$ is a predefined large number that $M \leq x$. The value of $h_i (1 \leq i \leq x)$ is 1 if $i = M$ else 0, and $r_i = N_i$. Only the first $M$ elements in $R$ have corresponding rate values, and other elements are all zero. Through number control from NE, when there are $M$ lines of Chinese texts in $T$, CLG is endowed with the ability to produce text layout $L$ with $M$ elements exactly. As each graphical character in the same text captures the same geometrical square space [11], introduced in Sec. 3.3, the size rate of the $i$-th element $w_i/h_i$ in $L$ must be equal to $i_N$. With RE, the origin DS-GAN generator can directly generate an estimated text layout $L^*$, satisfying $w_i/h_i$ very close to $i_N$. Then, PC transforms the $L^*$ to $L$ whose size rate of element completely matches the text length ($w_i/h_i = N_i$), which can be expressed as:

$$\text{PC}(L^*) = L = (l_1, l_2, ...l_M), \quad (6)$$

$$l_i^t = \begin{cases} (x_i^*, y_i^*, h_i^*, h_i^* \times N_i), \text{ if } w_i^*/h_i^* > N_i, \\ (x_i^*, y_i^*, w_i^*, w_i^*/N_i), \text{ else,} \end{cases} \quad (7)$$

where $x_i^*, y_i^*$ and $w_i^*, h_i^*$ is the estimated center location and size of the $i$-th Chinese graphical text. Upon processing by PC, the text layout $L$ thoroughly takes into account both the background image and Chinese text lines. This approach ensures the overall aesthetics and harmony of the final poster.

Figure 4: The overview of our Graphical Text Generator. The CLIP-based Text Attributes Predictor predict the color and font style embedding which is then used by the Stylized Font Generator to create the font. These elements collectively determine the Chinese graphical texts on the output poster.

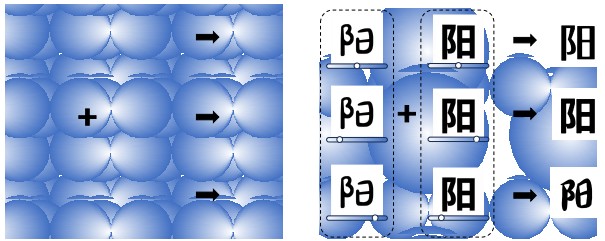

(a) Unlike pairs but same weight. (b) Same pair but unlike weights.

Figure 5: Interpolation of style embeddings. For the Chinese character, each graphical version corresponds to a style embedding in the feature space of the font style encoder in Diff-Font. So adjusting the weights of two styles can result in unlike graphical versions.

## 3.3 Graphical Text Generator

For an artistic Chinese poster, the correctness and style diversity of Chinese graphical texts are extremely crucial. The most critical weakness of previous stylized models is that they predict font in discrete space, leading to loss of font diversity [6, 22, 31]. Our proposed Graphical Text Generator (GTG) generates stylized Chinese graphical texts with diverse fonts and colors while guaranteeing their correctness. Thus, as the main improvement, we utilize the continuous feature space of Diff-Font [11] $\mathcal{G}_f$ to generate Chinese graphical texts character-wise, and each graphical Chinese character in the same text line captures the same geometrical square space. Specifically, $\mathcal{G}_f$ takes a font style encoder [34] $\mathcal{E}_f$ to create a continuous feature style space. Each style embedding in the space determines a specific font of the Chinese graphical character. For the same graphical character, any interpolation of two style embeddings brings a final font style in between, shown in Fig. 5.

Therefore, we can acquire Chinese graphical characters with unlimited font styles theoretically, by first predicting style embeddings $e$ and then getting the fonts with $\mathcal{G}_f$ based on $e$:

$$font = \mathcal{G}_f(e). \tag{8}$$

This highly enriches the font diversity compared with directly selecting limited font attributes in discrete space [6, 22, 31]. Following previous works [15, 31], the color prediction is directly in continuous pixel space. Noticing that $\mathcal{G}_f$ is a state-of-the-art font generation model, using which in our GTG naturally guarantees high correctness of generated Chinese graphical texts.

As shown in Fig. 4, the prediction of font attribute is achieved through modal alignment with $\mathcal{E}_f$, and the prediction of color attribute is accomplished through training from scratch. Specifically, the generation of the $i$-th font $g_i$ is determined by the Chinese text line $t_i$, background image $I$, and layout $l_i$. Since the local area has a strong influence on the style, we further obtain the local background $I_i$ based on $l_i$. Then, the pre-trained Text Encoder in Chinese CLIP [35] is used to encode $t_i$ to $f_i^t$, and Image Encoder in the same space is utilized to encode $I$ and $I_i$ to $f_i^I$ and $f_i^l$. Subsequently, we concatenate all the encoded features as the input feature $f_i = \text{concat}(f_i^t, f_i^I, f_i^l)$ for prediction.

To predict the corresponding style embedding $e_i$ and color value $o_i$ from concatenated feature $f_i$. We propose a Text Attributes Predictor (TAP), which incorporates a Feature Fusion Module (FFM) consisting of multiple MLPs, a linear Font Prediction Head ($\mathcal{H}_e$) and a linear Color Prediction Head ($\mathcal{H}_o$) for predicting the style embedding $e_i$ and color value $o_i$ respectively:

$$[e_i, o_i] = \text{TAP}(f_i) = [\mathcal{H}_e(\text{FFM}(f_i)), \mathcal{H}_o(\text{FFM}(f_i))]. \tag{9}$$

The font of the $i$-th Chinese graphical text $g_i$ is then obtained through $\mathcal{G}_f$ character-wise, based on their shared style embedding.

## 3.4 Loss Function and Optimization

We underline that the trainable module in Prompt2Poster is the Controllable Layout Generator (CLG) and the Graphical Text Generator (GTG). Both generators are supervised separately under corresponding losses, and combined with other proposed components as Prompt2Poster framework after training for automatic artistic Chinese poster creation.

The training of CLG is inherited from the basic DS-GAN [13], with a composition loss consisting of an adversarial component, a

reconstruction component, and general losses including NLL, L1, and IoU. For the training of GTG, two losses are introduced for supervising the font style and font color respectively. Noticing that the predictions for $i$-th line Chinese texts are style embedding $\boldsymbol{e}_i$ and color value $\boldsymbol{o}_i$. Concretely, we leverage the Kullback-Leibler (KL) divergence as the font loss $\mathcal{L}_{font}$ to distill the style distribution of real fonts into CTG, and $\mathcal{L}_{font}$ can be formulated as:

$$
\begin{aligned}
\mathcal{L}_{font} &= D_{KL}\left(\sigma(\boldsymbol{e}_i/t) \,||\, \sigma(\boldsymbol{e}_i^{gt}/t)\right) \\
&= \int \sigma(\boldsymbol{e}_i(x)/t) \log\left(\frac{\sigma(\boldsymbol{e}_i(x)/t)}{\sigma(\boldsymbol{e}_i^{gt}(x)/t)}\right) dx,
\end{aligned}
\tag{10}
$$

where $\boldsymbol{e}_i^{gt}$ is the ground-truth style embedding of $i$-th line, $\sigma(\cdot)$ is the softmax function and $t$ serves as the temperature factor to scale the feature distributions. In addition, we employ Mean Squared Error (MSE) as color value loss $\mathcal{L}_{color} = ||\boldsymbol{o}_i - \boldsymbol{o}_i^{gt}||^2$ to supervise the value between $\boldsymbol{o}_i$ and ground-truth $\boldsymbol{o}_i^{gt}$ directly.

Due to the significant difference in the scales of $\mathcal{L}_{font}$ and $\mathcal{L}_{color}$, we use uncertainty to balance multiple loss functions [17, 21], thus GTG is optimized as:

$$
\mathcal{L}_{GTG} = \frac{1}{2\gamma_1^2}\mathcal{L}_{font} + \frac{1}{2\gamma_2^2}\mathcal{L}_{color} + \mu\left[\log\left(1+\gamma_1\right) + \log\left(1+\gamma_2\right)\right],
\tag{11}
$$

where $\gamma_1$ and $\gamma_2$ are learnable parameters, while $\mu$ is the weight to scale the regularization term.

## 4 EXPERIMENTS

In this section, we present both quantitative and qualitative results to demonstrate proposed Prompt2Poster achieves superior artistic Chinese poster creation performance. Furthermore, extensive ablations are provided to certify the effectiveness of designed components in our Controllable Layout Generator and Graphical Text Generator. The implementation details are omitted due to the length limitation and can be found in our Appendix.

### 4.1 Comparison

We compared our Prompt2Poster with other five advanced prompt-guided poster generation methods, including SD XL [25], DALL·E 3 [1], Midjourney, GlyphControl [36], and SD XL + ControlNet (Canny) [37] for artistic Chinese poster creation. Following previous works [10, 22, 31, 32], we evaluate an artistic Chinese poster produced by the above methods from three aspects, including text quality, visual harmony, and overall attraction.

They are mainly determined by the visual correctness and style diversity of Chinese graphical texts, the matching between the background and Chinese graphical texts, and the overall artistic attraction to humans, respectively.

Since these artistic metrics are subjective, we conduct a user study to provide an intuitive quantification. Specifically, 41 people of various ages, gender, and occupation are invited to evaluate 30 groups' posters. Each group contains 6 posters which are generated by the above six methods from the same prompt. For the 6 posters in the same group, each participant compares and scores them in terms of the above three metrics respectively. It means that a poster will receive three different scores from a participant, corresponding to

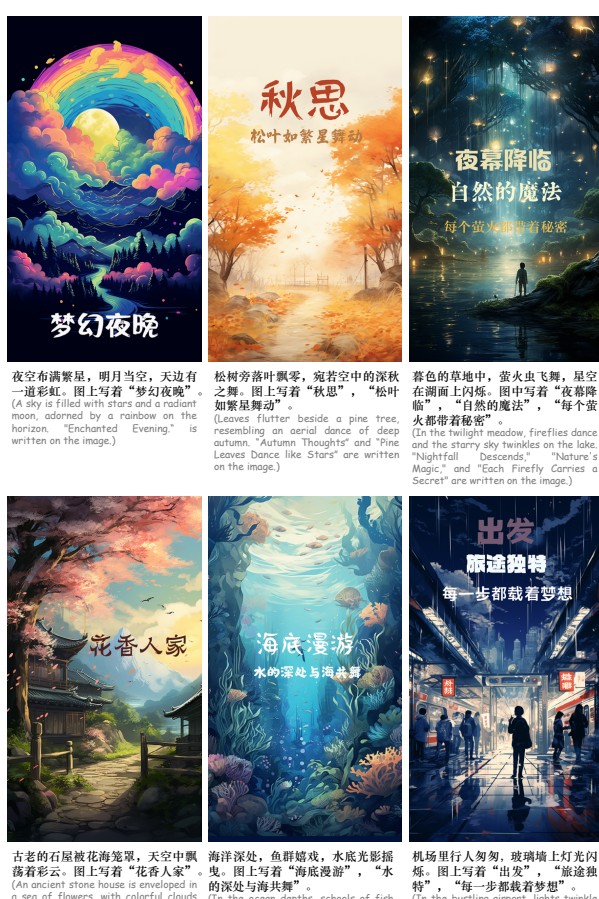

Figure 6: Prompt2Poster can generate artistic posters with Chinese texts in different row numbers.

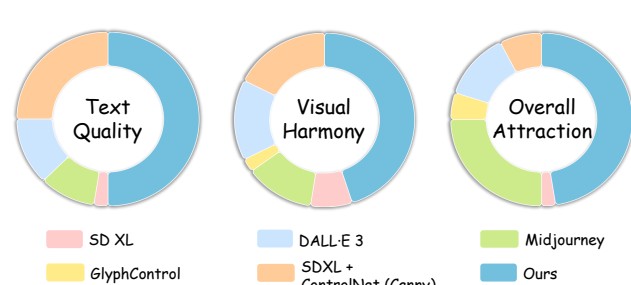

Figure 7: Pie chart results of user studies. Prompt2Poster achieves superior human preference in different aspects. Detailed values are provided in our Appendix.

the three metrics respectively. We set all the scores ranging from 0 to 10, and a higher score means the poster is better in the degree of a specific metric. Under the setting, we collect a total of 22140 ratings, bringing sufficient data support for the comparison. According to [10, 22, 31, 32], for each metric, instead of directly calculating the average scores for different methods, we statistic the times that the methods receive the highest score. The results are visually displayed

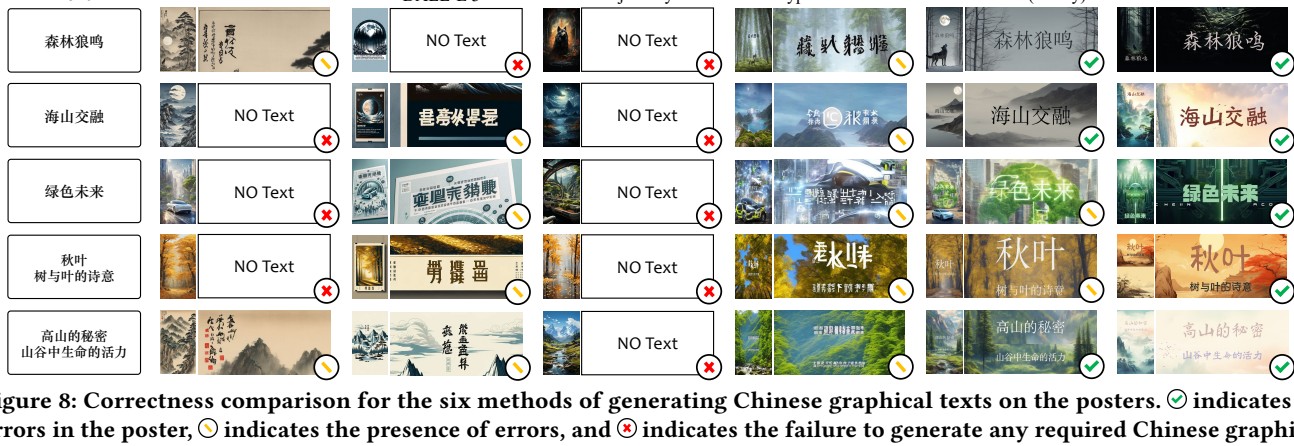

**Figure 8: Correctness comparison for the six methods of generating Chinese graphical texts on the posters.** ⊘ **indicates no errors in the poster,** ⊙ **indicates the presence of errors, and** ⊗ **indicates the failure to generate any required Chinese graphical texts. Except for our method, errors appear on all other methods and some of them even fail to generate graphical texts.**

through three pie charts, shown in Fig. 7. According to Fig. 7, for each metric, our Prompt2Poster all has a much higher frequency of receiving the highest score compared with any other methods. Concretely, for the occurrence percentage of achieving the best text quality and the best overall attraction, we are both nearly twice as much as the second-best method. For the occurrence percentage of achieving the best visual harmony, we are even close to three times as much as the second-best method. These indicate the great power of our Prompt2Poster for artistic Chinese poster generation. More details about the design of the user study are listed in the supplementary materials. Unlike other kinds of poster creation, the correctness of Chinese graphical texts is the foundation of artistic Chinese poster creation. Thus, a further comparison between our Prompt2Poster and other methods in terms of the correctness of Chinese graphical texts is conducted. Given some user prompts, all the methods generate corresponding posters and we only focus on the correctness of Chinese graphical texts, shown in Fig. 8.

According to Fig. 8, except for SD XL + ControlNet (Canny) and ours, the other four methods all fail to generate satisfying artistic Chinese posters in terms of correctness. Despite SD XL + ControlNet (Canny) having a better performance than the above four methods, it's still inferior to our Prompt2Poster which has a stronger correctness guarantee. These demonstrate that separately generating backgrounds and graphical texts from different distributions bring better fitting for the graphical data, enhancing the graphical correctness. As shown in Fig. 6, Prompt2Poster can generate posters with Chinese texts in different row numbers, incorporating elegant Chinese graphical texts. This is realized by a multi-modal framework design, which fully utilizes inner multi-modal information and completely exploits the potential of each module in the framework.

## 4.2 Ablation Study

**Controllable Layout Generator.** To verify the design effectiveness of our CLG, we also propose another two controllable text layout models, by removing modules in CLG. Specifically, we first remove module RE to get CLG (w/o RE), then SR-CLG is obtained

| Methods | Components | | Metrics (↓) | | | |
|---|---|---|---|---|---|---|
| | NE | RE | Ove | Ali | Occ | Rea |
| SR-CLG | | | 1.027 | **0.006** | 0.127 | 0.304 |
| CLG (w/o RE) | ✓ | | 0.017 | 0.010 | 0.115 | **0.301** |
| CLG | ✓ | ✓ | **0.004** | 0.008 | **0.108** | 0.302 |

**Table 1: Ablations for controllable layout generator with different components, where bold is the best performance.**

by removing RE and NE. Following [38], we utilize a self-regression strategy (SR) for SR-CLG to control the element number in the text Layout. PC is retained since it promises the text layout to finally match the length of the Chinese text. Training details for all these models are introduced in the supplementary materials.

To evaluate the text layout generated by the three models, We randomly selected 1500 background images and corresponding Chinese text lines as the input for them to generate specific text layouts. Following [13, 38], we compare the three models in terms of four numerical indexes, including Ove [18], Ali [2], Occ [13], and Rea [38]. These indexes measure the alignment, occlusion, and visual harmony of the text layout. For each index, the lower the value is, the better the layout is. The results are shown in Table. 1.

According to the Table. 1, taking self-regression strategy [38] without NE has a much higher Ove, which means the produced layout elements probably overlap together. This reflects the significance of the NE. Besides, the differences between Ali, Occ, and Rea among the three models are not obvious, indicating that adding modules for text control does not weaken other abilities. For a more intuitive visual comparison, we also run the original DS-GAN [13] by only providing the background images. Related results are shown in Fig. 9. According to Fig. 9, origin DS-GAN [13] generates text layouts with complete mistakes of number and size, leading to full incompatibility in our Prompt2Poster. Although promising the correct number of elements in $T$, SR-CLG faces serious element overlap, resulting in unacceptable results. Lacking RE, CLG (w/o RE) cannot estimate the final text layout in advance. This leads to significant cropping by PC and reduces space utilization. However,

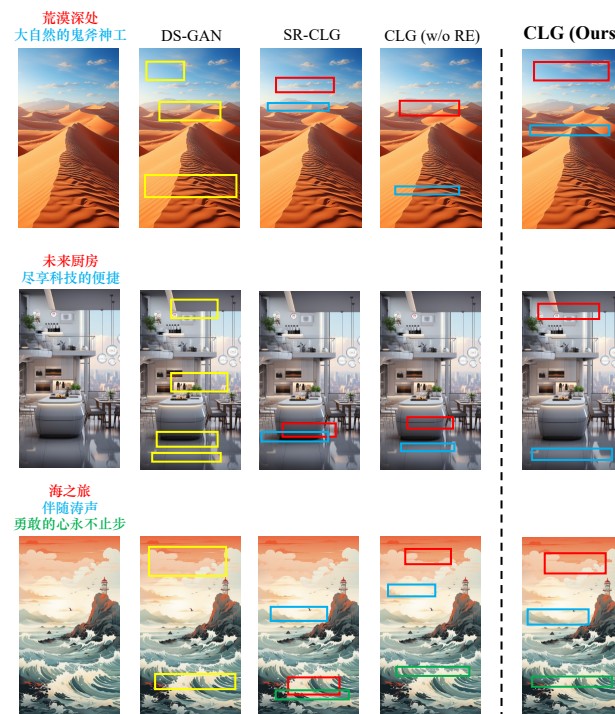

**Figure 9: Visual comparison among four layout models. The leftmost column presents input Chinese texts and background, and other columns show the text layouts generated by corresponding models respectively.**

with the addition of NE and RE, our CLG can produce a text layout that harmonizes with background and aligns texts accurately.

**Graphical Text Generator.** Besides keeping the visual correctness, the main purpose of utilizing GTG in our Prompt2Poster is to bring a huge style diversity of Chinese graphical texts, especially fonts. To verify that using GTG brings Prompt2Poster strongly stylized ability, we remove this module and obtain another system, named Prompt2Poster (w/o GTG). Specifically, following [37], we only exploit graphical templates to control the generation of Chinese graphical texts. The templates are created with the help of our CLG. More details are shown in the supplementary materials. We compared the style diversity of the Chinese graphical texts for Prompt2Poster (w/o GTG) and Prompt2Poster for each character, illustrated in Fig. 10. As Fig. 10 shows, without GTG, the styles of those generated Chinese graphical characters are much more monotonous. Particularly, although on different backgrounds, they seem to be very similar in terms of font style. With GTG, in terms of the graphical text style, our Prompt2Poster produces more diverse results and the fonts are much more elegant and varied.

In addition, it's worth mentioning that three modal inputs, including vision, geometry, and linguistics, all influence the output of our GTG. As shown in Fig. 11, any changes to the background image, layout, and Chinese text will bring variations in the generated Chinese graphical text. This full utilization of multi-modal information largely enhances the style diversity and harmony of the generated Chinese graphical texts.

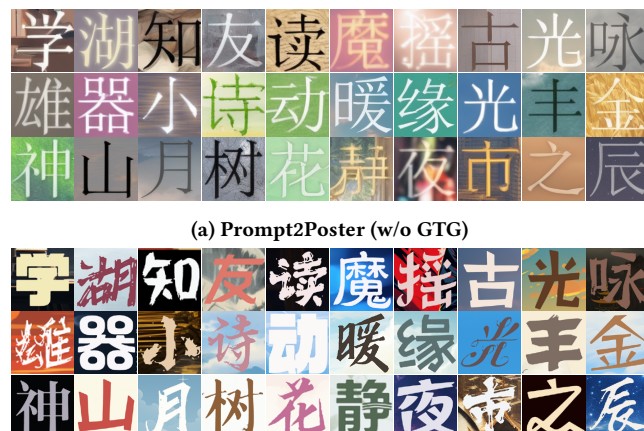

**Figure 10: Comparison of diversity in Chinese graphical texts between Prompt2Poster (w/o GTG) and Prompt2Poster.**

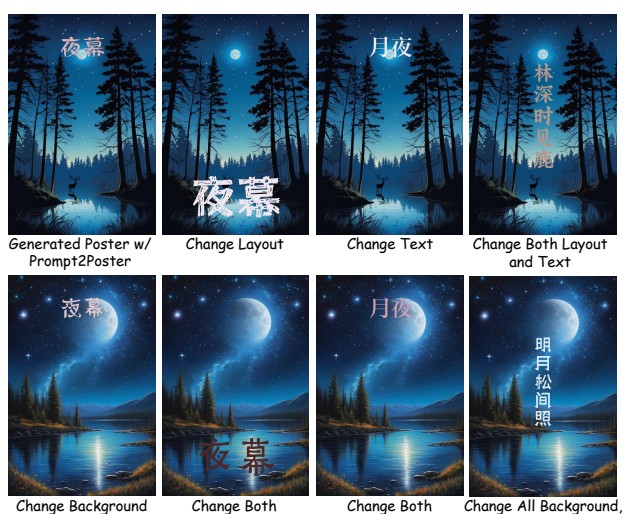

**Figure 11: Visualize ablations of different inputs of Graphical Text Generator (GTG), which demonstrate that GTG creates various graphical Chinese styles when inputs change.**

## 5 CONCLUSION

In this work, we propose an automatic prompt-guided poster creation framework named Prompt2Poster for creating desired artistic Chinese posters including an aligned background, reasonable layouts, and stylized graphical texts. Prompt2Poster first extracts the user intention and generates the aligned background supported by powerful large models, and then creates visually pleasurable content with our carefully designed modules including Controllable Layout Generator and Graphical Text Generator, leading to harmonious layout and accurate graphical texts. Extensive experiments demonstrate the superiority of our Prompt2Poster over other prompt-only poster generation methods. We believe our fully automatic poster creation framework will inspire potential research and interesting applications in the graphic design area.

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
