# OpenReview forum: "Prompt2Poster: Automatically Artistic Chinese Poster Creation from Prompt Only"
_acmmm.org/ACMMM/2024/Conference — MM2024 Poster_

### Official Review · Reviewer_fivi · 2024-05-22

**Rating:** 4
**Confidence:** 2

**Summary:**

Given that existing generative models either only output images or involve manual operations at certain steps for poster design, this paper focuses on poster generation based solely on a textual prompt as input. This approach is more convenient for users as it does not require them to provide a background image. The proposed model includes three modules: 1) A pre-trained Large Text-to-Image (T2I) Diffusion Model for background generation, where a large language model (LLM) is introduced to extract user intention from the provided prompts. 2) A Controllable Layout Generator (CLG) for layout generation conditioned on the generated background image. 3) A Graphical Text Generator (GTG) that generates text according to the prompt, layout, and background. The qualitative experimental results confirm the effectiveness of the proposed method.

**Strengths:**

1) The idea of generating a complete poster based on a prompt is novel and promising
2) This paper implements a comprehensive framework for poster generation by introducing diverse functions, which involves the whole poster design process.
3) The proposed evaluation metrics, including text quality, visual harmony, and overall attraction, are novel and reasonable. These metrics reflect a more subjective evaluation of the generated results.

**Limitations:**

1) How does the LLM extract texts expected to appear on the poster and divide the whole sentence into several text lines considering the line break in the font layout. In other words, how does it decide the number of lines (M)  for a text set (T).
2) The introduction of the trainable Number Encoder (NE), Rate Encoder (RE), and fixed Post-processing Cropping module (PC) needs to be clarified through a real-world case.
3) The Graphical Text Generator is complex and needs to be explained through equations. Additionally, for the CLIP-based predictor, it should be clarified whether the predicted color set and font style are chosen from a predefined set. If not, an explanation of how the prediction works is necessary.
4) The paper lacks quantitative experiments that compare the proposed model with SOTA (State of the Art) methods, which includes text to image generation, text+image to layout and font generation.
5) The experimental setup for both the baselines and the proposed framework should be clarified, including the models and datasets adopted for each part of the model or framework.

**Suitability:**

2

---

### Official Review · Reviewer_Q9CK · 2024-05-24

**Rating:** 4
**Confidence:** 4

**Summary:**

This paper aims to generate artistic posters based on the text prompt. The main contribution of this paper is the proposed pipeline for the artistic posters generation. Within this pipeline, a Controllable Text Layout Generator is proposed to generate text layout based on the background image and Chinese text, and a Graphical Text Generator is proposed to generate text with various colored fonts. The authors claim their model achieves superior performance than other existing methods, especially in text quality and visual harmony. The authors also claim that they will release their code after the paper review.

**Strengths:**

1. From the generated samples provided in this manuscript, the quality of the poster is promising.
2. This method can generate text images with various colors and fonts.
3. Based on the provided samples, the visual quality has significantly improved compared with other state-of-the-art methods.
4. This paper is well-prepared and well-written, making it easy to follow.

**Limitations:**

1. I notice the comparison in this paper mainly focus on the visualization comparison. The authors need to provide a quantitative comparison with other SOTA methods.
2. In this paper, the authors use SDXL, DALL E, Midjourney, GlyphControl, and SDXL+ControlNet for comparison, and only GlyphControl is designed for text generation. As the authors introduced in related works, there are some generative models focusing on text synthesis. I recommended the authors provide one other text generation model to verify their model.
3. Generating Chinese characters is difficult, since the number of characters is huge and the structure is complex. How many Chinese characters are supported in this model? What about the generative quality of uncommonly used Chinese characters?
4. What about the generation results for the characters with complex structures?
5. I notice the characters of the generated text only have a single color. Could this model generate text with the characters in different colors?
6. I suggest authors add a section to discuss the limitations of their model.

**Suitability:**

3

---

### Official Review · Reviewer_4RcC · 2024-05-24

**Rating:** 3
**Confidence:** 3

**Summary:**

This paper presents a method for generating Chinese posters based on prompts. The process begins with a Language Model (LLM) that extracts background descriptions and text content. Following the background description, a Text-to-Image (T2I) model creates the background image. Then, taking into account the background image and the text content, the layout for the texts are estimated. Finally, a graphical text generator predicts the text's color and font, which are combined with the background image through a font generation model. The paper introduces two multimodal models at the layout generation and text attribute estimation stages, called CLG and GTG, respectively.

**Strengths:**

This work effectively leverages a variety of recent models and displays impressive results.

**Limitations:**

- Limited technical novelty. Predicting the font and color of text has been explored in the work referenced in the paper [22], and the concept of predicting the embedding of the font, rather than the set of fonts itself, has also been addressed in existing research (Modeling Fonts in Context: Font Prediction on Web Designs, Pacific Graphics 2018).
- Lack of proper baselines for the whole pipeline, layout generator, and text generator.  For example, GlyphControl and SD XL GlyphControl with ControlNet could be fine-tuned on Chinese datasets before making comparisons, which is a very intuitive and fundamental approach.
- Lack quantitative metrics to assess the text generator's performance in terms of diversity, textual accuracy, harmony among multiple lines of text, and other aspects. From Figure 6, it is observed that within the same poster, the fonts and colors between different lines of text is not very harmonious, especially when there are three lines of text, resulting in significant variance in font styles between lines.
- Lack more quantitative analysis of the layout generator. For example, what impact do the number encoder and rate encoder have on the layout generation, and is the final layout consistent with the input conditions?

**Suitability:**

3

---

### Official Review · Reviewer_SQw9 · 2024-05-28

**Rating:** 4
**Confidence:** 3

**Summary:**

The paper presents Prompt2Poster, an automatic framework for creating artistic Chinese posters from user-provided prompts. Utilizing a Large Language Model (LLM), it extracts user intent to generate an aligned background and then employs a Controllable Layout Generator (CLG) and Graphical Text Generator (GTG) to produce harmonious layouts and stylized text. The work's main contributions include proposing a novel approach to artistic Chinese poster creation that leverages multi-modal information to accommodate different data distributions. The paper demonstrates superior performance in text quality and visual harmony compared to existing methods.

**Strengths:**

- Validity of Claims: Prompt2Poster is underpinned by large language models and advanced graphics generation techniques, offering a method for automatic poster creation. It demonstrates innovation and effectiveness in artistic poster creation through carefully designed modules.
- Importance and Novelty of Contributions: The work introduces a new framework for generating artistic Chinese posters, which is a novel contribution to the existing literature. It enhances the level of automation in poster generation and improves the visual and artistic impact of the posters.
- Relevance to ACM MM Community: The work is highly relevant to the research areas of the ACM MM community as it involves the creation, processing, and understanding of multimedia content, which are core technical issues in the multimedia field.

**Limitations:**

- Controllability Concern: The article transforms human control into an automatically generative model, and maintaining controllability in doing so is a concern.
- Robustness Concerns: The visualization results are all pictures of the landscape, there are no posters of the objects or movie posters.
- Comparative Analysis: The lack of comparison with the methodology of the related work section makes one more skeptical of its effectiveness.

**Suitability:**

2

---

### Meta-Review · Area_Chair_CgXM · 2024-07-04

**Recommendation:** Accept (Poster)
**Confidence:** 4

**Metareview:**

This paper presents a new approach to automatic poster creation leveraging a Large Language Model and advanced graphics generation techniques. The strengths of this work lie in its novel pipeline and the impressive visual results showcased. However, several limitations are also raised by the reviewers. These include providing quantitative comparisons with SOTA methods, clarifying certain technical components, and ensuring the controllability and diversity of the generated outputs.